# Ultrasound-Assisted Extraction of Flavonoids from *Potentilla fruticosa* L. Using Natural Deep Eutectic Solvents

**DOI:** 10.3390/molecules27185794

**Published:** 2022-09-07

**Authors:** He Xue, Jinping Li, Guiyao Wang, Wenming Zuo, Yang Zeng, Likuan Liu

**Affiliations:** 1College of Life Science, Qing Hai Normal University, Xining 810008, China; 2Academy of Plateau Science and Sustainable Development, Xining 810008, China

**Keywords:** natural deep eutectic solvents, ultrasound-assisted extraction, *Potentilla fruticosa* L., response surface methodology

## Abstract

A series of natural deep eutectic solvents (NADESs) were prepared with choline chloride, betaine, and a variety of natural organic acids in order to find new environmentally-friendly green solvents to replace the traditional solvents. The NADESs were employed to extract flavonoids from *Potentilla fruticosa* L. (*PFL*) with the help of ultrasound. The eutectic solvent diluted with an appropriate amount of water improved the extraction ability of flavonoids due to the decrease of solution viscosity. The microstructure of the raw sample and the samples subjected to ultrasonic bath in different solutions were observed using scanning electron microscope (SEM) to determine the role of the NADESs in the extraction process. The DPPH method and glucose consumption method were used to study the antioxidant and hypoglycemic ability of flavonoid compounds in *PFL*. Single factor method and response surface methodology (RSM) were designed to analyze the effects of three extraction parameters, including solvent/solid ratio, ultrasonic power, and extraction time, on the extraction yield, antioxidant capacity, and hypoglycemic capacity, and the corresponding second-order polynomial prediction models were established. The optimal extraction conditions for the maximum extraction yield, antioxidant capacity, and hypoglycemic capacity were predicted by RSM, and the reliability of RSM simulation results was verified by a one-off experiment.

## 1. Introduction

*Potentilla fruticosa* L. (*PFL*) is a typical deciduous shrub in the alpine region of China (such as mount Daban in Datong county, Qinghai province), belonging to the genus of *Potentilla* in the family *Rosaceae* [1]. In the theory of Traditional Chinese medicine, its leaves and flowers can be used as medicine, having the functions of clearing the summer heat, nourishing the brain, and clearing the heart, as well as regulating menstruation and invigorating the stomach [2,3]. The biological activity of *PFL* stems from the fact that it contains a variety of flavonoids, terpenoids, and tannins, among which flavonoids are the most typical substances [4,5]. Flavonoid aglycones mainly include apigenin, quercetin, kaempferol, luteolin, rhamnetin, isorhamnetin, myricetin, and so on, which have antihypertensive, hypolipidemic, and antioxidative effects [6]. Therefore, the extraction and separation of the active ingredients in *PFL* have special significance in medicine, food, and daily chemicals. In fact, the natural compounds of different species all have very important potential applications and have attracted the attention of many researchers in recent years [7,8,9].

Cold leaching extraction, decocting extraction, hot reflux extraction, and Soxhlet extraction are the conventional methods for extracting active substances from plant materials [10]. Generally speaking, the extraction rate of the cold leaching extraction is not high, while other methods often require higher extraction temperature, which may damage the temperature-sensitive components of the active substances [11,12], such as volatile essential oils and carotenoids, which are particularly sensitive to light and heat and thus are not suitable for high temperature extraction. Ultrasonic extraction is an efficient, safe, and inexpensive extraction technology. The strong tensile force and cavitation effect produced by the ultrasound breaks the plant cell wall and the active substances are more easily released into the extraction solvent [13]. Therefore, ultrasonic extraction technology is widely used in the extraction field of plant active substances [14,15,16]. Safety and environmental considerations of the extraction technology are particularly important in the pharmaceutical, food, and cosmetic industries, however, volatile organic solvents (VOSs) are usually used as extraction agents in traditional extraction processes, such as cold quenching extraction and thermal reflux extraction. VOSs are usually non-environmentally friendly and have certain biological toxicity, which may cause great damage to the health of the experimenter due to the strong volatility. Meanwhile, the flammable and explosive characteristics of VOSs also bring hidden dangers in production safety [17]. Therefore, safe and environmentally-friendly extraction solvents have always received the attention of pharmaceutical, food, and cosmetic industries.

Deep eutectic solvents (DESs) are a class of green and sustainable solvents which were first introduced by Abbott et al. in 2003 [18]. A DES generally can be prepared by mixing two or more solid or liquid components together according to a certain mole ratio, in which one acts as the hydrogen bond donor (HBD) and the other as the hydrogen bond acceptor (HBA) [19,20]. The HBD and HBA can form intermolecular hydrogen bonds and be bound together by van der Waals interaction [21,22]. DESs can be used as favorable substitutes of conventional VOSs in pharmaceuticals, extractions, catalysis, and other fields, owing to their many advantageous properties, such as low volatility, low toxicity, being non-flammable, high relative solubility, and ease of synthesis [23,24]. Although the components of DESs are non-toxic and environmentally friendly, there is no guarantee that DESs themselves will have the same properties. Hayyan et al. [25,26] found that the cytotoxicity of four ordinary DESs prepared from glycerol, ethylene glycol, triethylene glycol, urea, and choline chloride was much higher than that of their components, which indicates that the common DESs still have low biotoxicity. Therefore, with the development of DESs, more and more attention has been paid to the natural deep eutectic solvents (NADESs) composed of natural plant components such as choline chloride, different sugars, organic acids, and phenols. NADESs are basically completely non-toxic, and have better biocompatibility and high solubility of natural products compared with the ordinary DESs [27].

In this study, two natural organic bases (choline chloride and betaine) were used as the HBA and four natural organic acids (malic acid, tartaric acid, citric acid, and proline) were used as the HBD to prepare different NADESs. Using these NADESs, flavonoids in *PFL* were extracted by the ultrasonic method. The effects of solid/liquid ratio, concentration of NADESs, ultrasonic power, and ultrasonic time on the extraction rate of flavonoids were systematically studied. Response surface methodology (RSM) was used to optimize the extraction parameters, and the optimal extraction parameters were obtained. Multiple series column chromatography was used to separate and purify flavonoids from *Potentilla rubra*. DPPH analysis was used to evaluate the antioxidant activity of the isolated flavonoids; the hypoglycemic activity of the flavonoids was also tested, which provided technical support for the extraction and high-value utilization of flavonoids from *PFL*.

## 2. Experimental Methods

### 2.1. Chemicals and Plant Material

*PFL* was picked from Daban Mountain, Datong County, Qinghai Province (altitude 3006 m, 37°8′42″ N, 101°50′32″ E). The stems and leaves of *PFL* were completely dried at 40° and then crushed into powder with a particle size of 80 mesh.

Two organic bases and three organic acids with purities >99 wt%, namely choline chloride, betaine, citric acid, malic acid, DL-tartaric acid, and L-proline, were purchased from Qiansheng biotechnology Co., Ltd. (Hefei, China). Rutin standard with purity >99 wt% was purchased from Ruifansi biological technology Co., Ltd. (Chengdu, China). 1, 1-diphenyl-2-trinitrophenylhydrazine (DPPH), Dimethyl sulfoxide(DMSO) with purities >99.5 wt% were purchased from Aladdin biochemical technology Co., Ltd. (Shanghai, China). 3T3-L1 cells, insulin, and glucose were purchased from Yipu biotechnology Co., Ltd. (Wuhan, China).

### 2.2. Synthesis of NADESs

As shown in Table 1, eight kinds of DESs were prepared with choline chloride and betaine as the HBA and a variety of organic acids as the HBD. According to the mole ratio shown in Table 1, the two components were mixed and fully dissolved by adding appropriate amounts of water, then put into a sealed conical flask. The conical flask was placed on a S10-3 thermostatic magnetic stirrer (Shanghai Sile Instrument Co., Ltd., Shanghai, China) and magnetically stirred at 40 °C for 24 h. The excess water in the obtained transparent liquid was dried in a C-2L-MRE rotating evaporator (Shanghai Sheyan Instrument Co., Ltd., Shanghai, China) to obtain the transparent and uniform eutectic solvent. In liquids with high viscosity, the mass transfer behavior of the active substance will be significantly delayed, which could obviously impair the extraction efficiency of the extraction solvent. Due to high viscosity of the eutectic solvents, they should be properly diluted to facilitate the extraction tests.

### 2.3. Ultrasonic Extraction and Microstructure of Plant Material

0.5 g powder of *PFL* was accurately weighed and placed in a 20 mL flask. The DES sample was diluted with water to a specific concentration before extraction, then added to the flask according to a certain solid/liquid ratio. The ultrasonic emission probe of a Scientz-IIDZ ultrasonic cell crusher (Ning Bo Xinzhi Biotechnology Co., Ltd., Ningbo, China) was inserted into the mixture, and the ultrasonic input power and time were set to carry out ultrasonic extraction of the mixture. The extracted solution was put into a TG16MW high-speed centrifuge (Hunan Husi instrument equipment Co., Ltd., Changsha, China), and centrifuged at a speed of 8000 r/min for 15 min. The supernatant in the centrifuge tube was then filtered through a 0.45 μm PTFE syringe filter for high performance liquid chromatography (HPLC) analysis. Extraction under each condition was repeated three times to ascertain reliability, and the average value was reported.

The extraction yield was defined by Equation (1):(1)$$Y=\frac{m_{1}}{m_{0}}\times 100\%$$
where *Y* is the extraction yield (mg/g), *m*_1_ is the mass of flavonoids extracted in the solution (mg), and *m*_0_ is the mass of the *PFL* powder added (g).

### 2.4. HPLC Analysis

The concentrations of *PFL* extracted by DES solutions were determined using an Agilent 1200 series HPLC equipped with a quaternary pump, a DAD detector, an auto sampler, and a thermostatted column compartment. An Intersil ODS-3C18 column (250 mm × 4.6 mm, 5 μm) was used to detect chromatographic separations. The mobile phase was prepared with methanol and water, while a gradient elution program was performed with the methanol solution (the eluent was progressively increased from a volume concentration of 10% to 100% in increments of 10% per step) with a flow rate of 1.0 mL/min for 10 min at each stage. Another 10 min post-run time was carried out to fully equilibrate the system. The column oven was held at 25 °C while the UV spectra were controlled at 360 nm. The sample injection volume was 20 μL.

The calibration curves for *(+)-catechin-7-O-glucoside*, *(+)-catechin, quercetin-3-O-β-D-glucuronide*, and *quercetin 7-O-β-D-glucuronide* were: *Y* = 38.236*X* − 0.0142 (*R* = 0.9999) for *(+)-catechin-7-O-glucoside*, *Y* = 24.365*X −* 0.0237(*R* = 0.9999) for *(+)-catechin*, *Y* = 18.438*X* − 0.0312 (*R* = 0.9999) for *quercetin-3-O-β-D-glucuronide*, and *Y* = 49.652*X −* 0.0136 (*R* = 0.9999) for *quercetin 7-O-β-D-glucuronide*, where *X* was their amount (mg) and *Y* was the UV absorbance value of their corresponding peak area [28].

### 2.5. Response Surface Methodology

Response surface methodology (RSM) is a statistical technique for investigating the interactions between factors in an appropriate variable range [28,29]. The effects of the solvent/solid ratio, ultrasonic power, and extraction time were investigated by RSM to obtain the suitable extraction yield of flavonoids with optimal antioxidant capacity and hypoglycemic capacity. The Box-Behnken design (BBD) combined with RSM was applied to three independent variables, including (A) solvent/solid ratio, (B) ultrasonic power, and (C) extraction time at five levels (−a, −1, 0, +1, and +a) to study their relation to the extraction yield, the antioxidant ability, and hypoglycemic capacity. The Design-Expert 10.0 software package (Stat-Ease Inc., Minneapolis, MN, USA) was used for the generation and evaluation of the experimental design.

### 2.6. Determination of Antioxidant Capacity

DPPH analysis is a widely used method for screening free radical scavengers [30]. The antioxidant activity of flavonoids extracted in the prepared DES solutions was evaluated using the DPPH analysis method. A certain amount of DPPH was weighed and prepared into 0.04 mg/mL DPPH solution with anhydrous ethanol. Different concentrations of sample solutions (2, 4, 6, 8 mg/mL) were prepared with anhydrous ethanol. 2 mL DPPH solution was added to 2 mL sample solution, and the mixture was evenly mixed and then left to react for 30 min at room temperature. The mixed solution was centrifuged at 5000 r/min for 10 min, and the absorbance value of the supernatant was measured at 517 nm. Anhydrous ethanol was used as a blank control group. The antioxidant activity of the samples was calculated according to Equation (2):(2)$$C=\frac{A_{0}-\left(A_{S}-A_{C}\right)}{A_{0}}\times 100\%$$
where, *C* is the DPPH free radical scavenging rate (%); *A*_0_ is the absorbance of the test sample of 2 mL deionized water and 2 mL DPPH solution; *A_s_* is the absorbance of the test sample of 2 mL sample solution and 2 mL DPPH solution; *A_c_* is the absorbance of the test sample of 2 mL sample solution and 2 mL anhydrous ethanol.

### 2.7. Determination of Hypoglycemic Capacity

The 3T3-L1 cells thawed in the water bath were centrifuged and evenly added to the petri dish containing fresh complete medium by blowing and beating. The cells were cultured at 37 °C in a 5% CO_2_ incubator, and the culture medium was replaced every other day. The passage was carried out when the cells reached logarithmic stage. After the passage cells were overgrown, 3T3-L1 preadipocytes were inoculated into 24-well plates at a density of 10^5^/mL, and cultured in an incubator with 5% CO_2_ at 37 °C until the cells were fully fused, and then induced differentiation was conducted.

The specific method of inducing differentiation was as follows: Inducer I (complete medium containing 10 μg/mL insulin, 1 μmol/L Dex and 0.5 mmol/L IBMX) was added to the culture for 48 h and then Inducer II (complete medium containing only 10 μg/mL insulin) was added to the culture for 48 h. The complete medium was changed every 1 to 2 days for further culturing. After differentiation to the 8th generation, the cells were washed with 1800 mg/L glucose and added to 200 μL low-glucose medium for incubation. The sample groups to be tested were incubated with 25 μM final concentration flavonoid extracts, and the control groups were incubated with DMSO and 100 nM insulin, respectively. Each sample was set with 3 repeat wells. After incubation for 24 h, 10 μL cell culture medium was extracted and the concentration of glucose in the culture medium was determined by the glucose oxidase-peroxidase method.

Glucose concentration was calculated by Equation (3):(3)$$C=\frac{A_{S}\times C_{c}}{A_{st}}$$
where, *C* is the glucose concentration (mmol/L), *A_s_* is the absorbance of the samples, *C_c_* is the concentration of the calibration product (mmol/L), *A_st_* is the absorbance of the standard product.

Glucose consumption rate was defined by Equation (4):(4)$$P=\frac{C_{0}-C_{t}}{C_{0}}\times 100\%$$
where, *P* is the glucose consumption rate (%), *C_0_* is the initial glucose concentration, and *C_t_* is the glucose concentration in the experimental well.

## 3. Results and Discussion

### 3.1. Preparation and Physicochemical Properties of DES

Studies have shown that the viscosity of DESs diluted with water is significantly reduced, and the quality of the solvent is also obviously improved [31,32]. Table 1 displays the viscosity and PH values of 70 *v*/*v*% DES aqueous solutions; it can be seen that the viscosities of diluted DES aqueous solutions were in the range of 1.586~1.966 mPa*s, only slightly higher than ethanol (1.11 mPa*s) and water (0.9 mPa*s), which fully meets the requirements for the liquid viscosity less than 100 mPa*s in industrial applications [33]. The low viscosity of the DES aqueous solutions could lead to high diffusivity, which is beneficial to improve the extraction performance [34]. The pH value is another factor that affects the extraction efficiency. The pH values of the diluted DESs were between 1.57~1.82, which indicated they had strong acidity.

### 3.2. Extraction of Flavonoids Using DES Aqueous Solution

In order to compare the extraction effects of different solvents, eight DESs and two control solvents were employed to extract flavonoid compounds from *PFL* under the same ultrasound-assisted conditions (solvent/solid ratio of 30/1, 70 *v*/*v*% aqueous solutions of DES, 300 W of ultrasonic power, 30 min of ultrasonic extraction time, 28 °C of environment temperature). Figure 1 displays the extraction yield of four flavonoid compounds ((+)-catechin-*7-O*-glucoside, (+)-catechin, quercetin-3-*O-β*-D-glucuronide, and quercetin 7-*O-β*-D-glucuronide) in *PFL*. As can be seen from Figure 1, the extraction yields of flavonoids in DESs were significantly higher than that in ethanol and water, which indicated that DESs have better extraction capacity than the traditional solvents (ethanol and water), and the flavonoid contents in diluted DESs were mainly attributed to DESs themselves. That is in agreement with many extraction studies of natural products with DESs [35,36,37]. Tukey’s test was used to compare the difference of flavonoid extraction yield between DES-3 and DES-7 solutions, which were the most similar in extraction yield values. The final count of the two extraction rate data was eight, which was greater than the critical value of the final count of seven at the 95% confidence level, indicating significant variation between the flavonoid extraction yield of the two solutions. DES-3, synthesized from choline chloride and citric acid, presented the highest extraction efficiency compared with other DESs. It afforded (+)-catechin-7-*O*-glucoside, (+)-catechin, quercetin-3-*O-β*-D-glucuronide, and quercetin 7-*O-β*-D-glucuronide in the yields of 12.2, 5.4, 7.2, and 17.7 mg/g, respectively. For binary DESs, the choline chloride series of DESs (DES-1, DES-2, DES-3) exhibited better extraction capability of flavonoids from *PFL* than the betaine series of DESs (DES-5, DES-6, DES-7), while ternary DESs (DES-4 and DES-8) showed opposite extraction abilities. The extraction capacity of DESs for natural products depends on the types of HBA and HBD in DESs and the molar ratio between them [38]. The differences in extraction performance of the DESs studied in this paper were related to the multiple interactions between DESs and flavonoid compounds, including ionic/charge–charge and hydrogen bonding interactions [39]. However, there is no clear explanation to elaborate on the superior extraction performance of some DESs than others.

Figure 2 displays the microstructure of the powder granules of *PFL* before and after ultrasonic treatment in water, ethanol, and 70 *v*/*v*% aqueous solution of DES-3 baths. As shown in Figure 2a, the raw powder had a flat and mostly smooth surface with little roughness in local areas. In contrast, the surface of the powder treated with water and ethanol gradually became rough and coarse, with increasing irregular folds generated (Figure 2b,c). After 70 *v*/*v*% aqueous solution of DES-3 treatment, the surface of the powder appeared to deteriorate obviously, evidenced by the appearance of a large number of voids, as shown in Figure 2d. This may be due to the effective dissolution effect of DES on cellulose [40,41], coupled with the cavitation effect and mechanical vibration caused by ultrasonic radiation [42], resulting in the damage of plant cell structure, which allowed the target flavonoid compounds to be more easily dissolved in the extraction solvent. This is in agreement with the extraction yield reported in Figure 1, which showed higher flavonoid content for DES-3 compared with water and ethanol.

### 3.3. Effects of Water Content and Ultrasound-Assisted Extraction Conditions

In order to investigate the effect of different water content on the flavonoid extraction yield of DESs, three DESs (DES-2,3,7) with higher extraction yields were selected for further experimental study according to the experimental results shown in Figure 1. Figure 3 shows the extraction yield of flavonoids for three DESs diluted into different concentrations. Compared with the DESs with other concentrations, the 70 *v*/*v*% of DESs exhibited the best extraction performance. The better extraction performance of DES diluted by appropriate amounts of water can be attributed to the better mass transfer behavior caused by the decrease of solution viscosity [43]. However, the addition of excessive water will not only reduce the solubility of flavonoid compounds in the extraction solutions, but also affect hydrogen bond interactions between the components of the DESs [44], which may be the main reason for the change of the physicochemical properties of the diluted DESs.

As mentioned in the previous section, DES-3 was employed to optimize the ultrasonic extraction conditions due to it showing the best extraction performance compared with the other DESs. Figure 4 illustrates the extraction yield of flavonoids in the 70 *v*/*v*% DES-3 at different solvent/solid ratios, ultrasonic power, and extraction time. As shown in Figure 4a, when the solvent/solid ratio increased from 10:1 to 30:1, the extraction yield of flavonoids increased significantly, but the increasing trend became unobvious when the solvent/solid ratio continued to increase to 50:1. For a certain quality of *PFL* materials, an appropriate increase in the amount of extraction solution was conducive to better dissolution of flavonoids in the materials. However, the dissolution demand of limited flavonoids in the materials had been basically met when the extraction solvent reached a certain amount. Even if the extraction solvent content continued to increase, not only the dissolution increment of flavonoids would become relatively small, but also the extraction cost would increase. The data presented in Figure 4b reveal the enhancement of the amounts of the extracted flavonoids with increasing ultrasonic power up to 300 W, while the opposite trend appeared when the ultrasonic power continued to be increased. Ultrasonic waves can produce strong cavitation, mechanical vibration, and disturbance effects, which can destroy the tissue structure of plant cells [45], resulting in rapid dissolution of target flavonoid compounds in the cells of *PFL.* However, the extraction effect cannot always be enhanced with the increase of ultrasonic power because the bond structure of flavonoid compounds may be destroyed by high ultrasonic energy, resulting in the reduction of flavonoid extraction content [46]. The extraction time was also an important factor affecting the extraction yield of flavonoids. Figure 4c demonstrates that the extraction yield of flavonoids maintained a significant trend of increase until the extraction time reached 40 min, but there was only a small growth in yield when the extraction time ranged from 40 to 60 min, which indicated that the dissolution of flavonoids had been completed within 40 min. In summary, considering saving extraction time and cost, the preferred ultrasonic extraction parameters were determined to be a solvent/solid ratio of 30:1 mL/g, ultrasonic power of 300 W, and an extraction time of 40 min.

### 3.4. Response Surface Methodology

Based on the three single factor experiments in the previous section, the effects of the three independent variables, the solvent/solid ratio (A: 20:1–40:1 mL/g), ultrasonic power (B: 200–400 W), and extraction time (C: 30–50 min), were investigated by RSM to obtain the optimal ultrasonic extraction conditions. The antioxidant assay and hypoglycemic assay were used to evaluate the quality of flavonoids extracted in the DES solutions. The 70 *v*/*v*% DES-3 solution was employed for further response surface analysis as it had been demonstrated as one of the most effective extractants among the prepared DESs. Table 2 displays 17 runs of BBD experiments and their results, which were used to further optimize the extraction conditions, antioxidant capacity, and hypoglycemic capacity of flavonoids. The greatest extraction yield was obtained at the 7th run and the largest DPPH clearance rate and glucose consumption rate were also obtained at the same run. The minimum extraction yield was obtained at the 9th run and the smallest DPPH clearance rate and glucose consumption rate were also obtained at the same run. This indicated that the effect of extraction parameters on extraction yield was nearly consistent with the effect on antioxidant and hypoglycemic performance. As can be seen from the single factor experiments, although the content of total flavonoids was different under different extraction conditions, there was no significant difference in the content proportion of four compounds in the extract. Since all extraction experiments were carried out at room temperature, the activity of each compound would not be affected by temperature [47]. Therefore, the antioxidant capacity and hypoglycemic capacity of the total flavonoids were only related to the total content, and were not affected by the differences of the compound types. The highest flavonoid extraction yield obtained from the BBD matrix experiments was basically consistent with the previous single factor experiments, indicating that the recommended ultrasonic extraction parameters were liquid–solid ratio of 40:1 mL/g, ultrasonic power of 300 W, and ultrasonic time of 50 min.

The simulations of the experimental data by the Design-Expert 10.0 software package could predict the extraction yield of flavonoids, DPPH scavenging rate, and glucose consumption rate via three second-order polynomial equations, which were as follows:Yield = 47.50 + 2.99 ∗ A + 1.55 ∗ B + 3.63 ∗ C + 0.20 ∗ A ∗ B − 0.32 ∗ A ∗ C + 0.013 ∗ B ∗ C − 1.37 ∗ A^2^ − 4.94 ∗ B^2^ − 2.59 ∗ C^2^
DPPH scavenging rate = 51.40 + 3.28 ∗ A + 1.66 ∗ B + 3.96 ∗ C + 0.25 ∗ A ∗ B − 0.35 ∗ A ∗ C + 0.025 ∗ B ∗ C − 1.44 ∗ A^2^ − 5.36 ∗ B^2^ − 2.81 ∗ C^2^
Glucose consumption rate = 17.18 + 1.10 ∗ A + 0.56 ∗ B + 1.34 ∗ C + 0.075 ∗ A ∗ B − 0.075 ∗ A ∗ C + 0.000 ∗ B ∗ C − 0.49 ∗ A^2^ − 1.82 ∗ B^2^ − 0.92 ∗ C^2^

The 3D response surface plots and contour plots in Figure 5 illustrate the interaction effects between the variables affecting the extraction yield of flavonoids, DPPH scavenging rate, and glucose consumption rate. The highest yield of flavonoids for the ultrasonic power appeared at between 300 W and 350 W in Figure 5a,c whereas both extended liquid–solid ratio and extraction time increased the extraction efficiency, as seen from Figure 5b. It can be seen from the elliptic degree of contour lines that there were obvious interactions between liquid–solid ratio and ultrasonic power, as well as ultrasonic power and extraction time. The extraction solution can break through the barrier of the cell wall more easily due to the damage of cell structure caused by the ultrasound combined with the erosion of cellulose by DES solvent, so that more target compounds were dissolved in the extraction solution [48]. However, higher ultrasonic power could damage the binding structure of the target compounds, which inhibited the further improvement of extraction efficiency [49]. Moreover, this negative effect would become more severe as the extraction time increased. As seen from Figure 5d–i, the interaction of the extraction variables on DPPH clearance rate and glucose consumption rate were basically the same as that on extraction yield. The optimal extraction conditions from the simulation for obtaining reasonable content of the flavonoids and the maximum DPPH clearance rate and glucose consumption rate were 40:1 mL/g of solvent/solid ratio, 317.6 W of ultrasonic power, and 46.9 min of extraction time. To examine the calculated optimal parameters, a one-off experiment was performed with 70 *v*/*v*% DES-3 with a solvent/solid ratio of 40:1 mL/g at 318 W ultrasonic power for 47 min. A slight difference was observed between the predicted extraction yield (50.33 mg/g) and the experimental result (50.26 mg/g), along with similar antioxidant assay results (DPPH clearance rate: 54.6% vs. 54.3%) and hypoglycemic assay results (glucose consumption rate: 18.3% vs. 18.1%), which suggested that the prediction model established by the RSM approach could accurately and efficiently obtain the optimal technological conditions. In fact, the RSM approach has been widely used in the extraction field of active substance, and has been verified by many researchers with high precision and high efficiency [50,51,52].

## 4. Conclusions

Eight NADESs were successfully prepared and used to extract flavonoids from *PFL* under ultrasonic-assisted conditions. The extraction ability of NADESs was significantly higher than that of traditional solvents, and the choline-citric acid eutectic solvent showed the best extraction performance for flavonoids. The changes in the microstructures of the treated *PFL* powders demonstrated that DES solution broke the cell structure more effectively and promoted the extraction solution to break through the barrier of the cell walls with the help of ultrasound. The extraction capacity of DESs can be further improved when diluted with appropriate water, which is attributed to the reduction of solution viscosity, promoting the mass transfer of the active substance. The optimal dilution concentration was 70 *v**/**v*%. Ultrasonic power is an important parameter affecting the extraction yield of flavonoids. Increasing the ultrasonic power can significantly increase the extraction yield, but excessive ultrasonic energy will produce the completely opposite effect. Increasing solvent/solid ratio and extraction time promoted the increase of extraction yield, but when they reached a certain level, the increase of extraction yield brought by further increase became very limited. The interaction of the extraction variables on DPPH clearance rate and glucose consumption rate were basically the same as that on extraction yield. Using RSM and a one-off validation experiment, the optimal extraction conditions with the maximum extraction yield, antioxidant ability, and hypoglycemic ability were determined to be 40:1 mL/g of solvent/solid ratio, 318 W of ultrasonic power, and 47 min of extraction time. There was a slight difference between the predicted values and the experimental values, confirming that RSM was a reliable and efficient method for determining the optimal extraction conditions. This study provided an example on the use of NADES for replacing traditional solvents and enabling efficient extraction. Aqueous solutions of DES have the potential to be safely used in cosmetic, pharmaceutical, and nutraceutical applications.

## Figures and Tables

**Figure 1 molecules-27-05794-f001:**
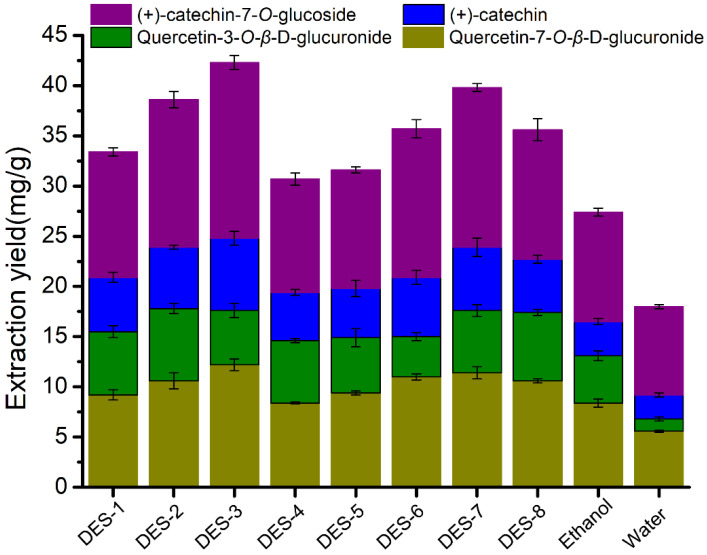
Extraction yield of flavonoids in 8 different 70 *v*/*v*% aqueous solutions of DES, ethanol, and water after ultrasonication-assisted extraction determined by HPLC analyses.

**Figure 2 molecules-27-05794-f002:**
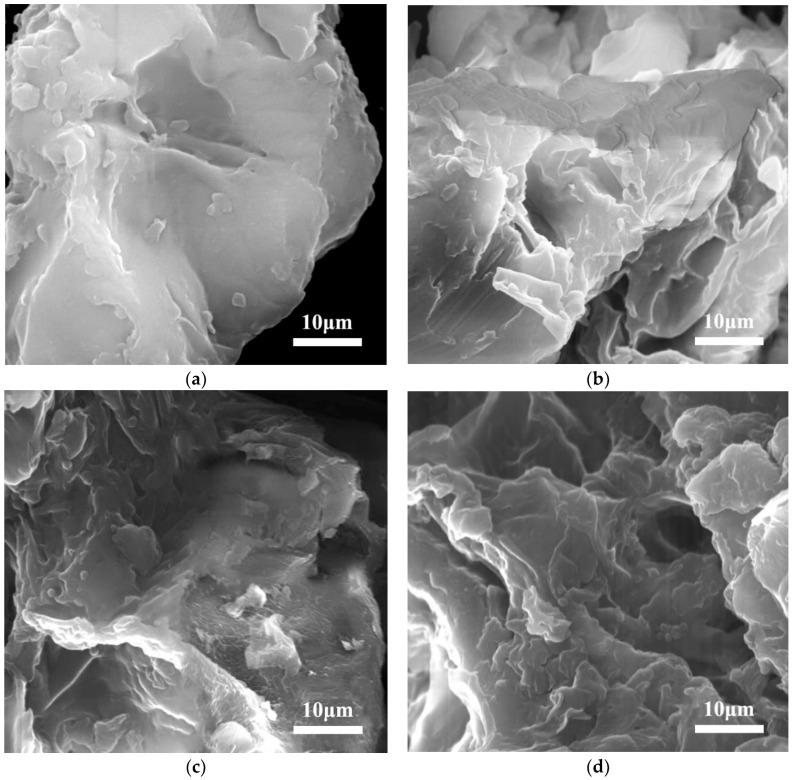
Morphological photos of the *PFL* powders (**a**) before ultrasonic treatment and after ultrasonic extraction in (**b**) water, (**c**) ethanol, and (**d**) 70 *v*/*v*% aqueous solution of DES-3 baths with a solvent/solid ratio of 30/1 at 28 °C for 30 min.

**Figure 3 molecules-27-05794-f003:**
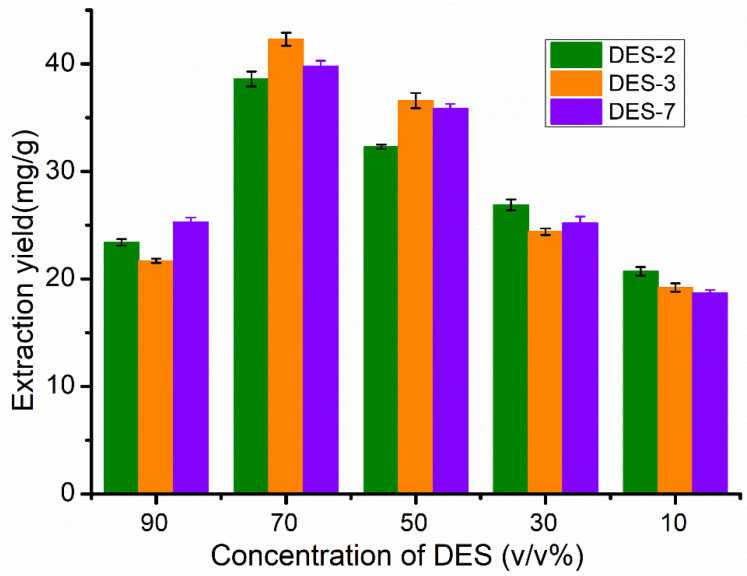
Extraction yield of flavonoids in the three DESs (DES-2,3,7) with different concentrations via ultrasound-assisted extraction with solvent/solid ratio of 30/1 at 28 °C for 30 min.

**Figure 4 molecules-27-05794-f004:**
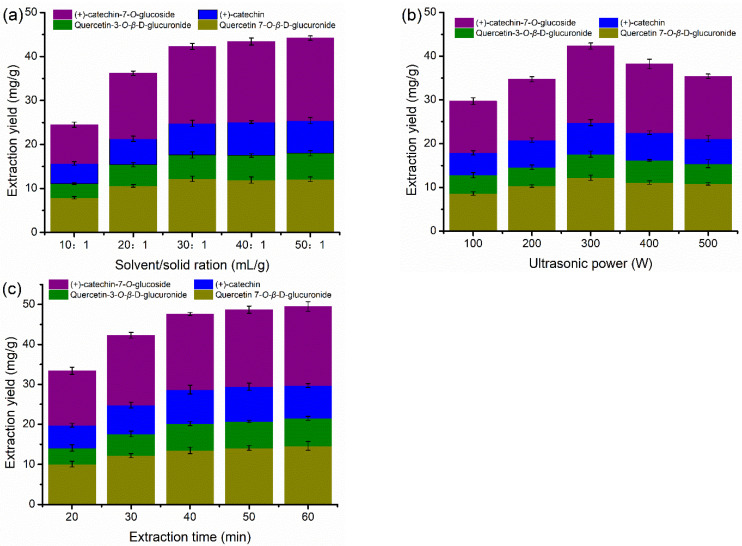
Conditions for extraction and yield of flavonoids in the 70 *v*/*v*% DES-3 ultrasonic bath. (**a**) effect of solvent/solid ratio at ultrasonic power of 300 W for 30 min. (**b**) Effect of ultrasonic power with solvent/solid ratio in 30/1 mL/g for 30 min, and (**c**) effect of extraction time with solvent/solid ratio in 30/1 mL/g and at ultrasonic power of 300 W.

**Figure 5 molecules-27-05794-f005:**
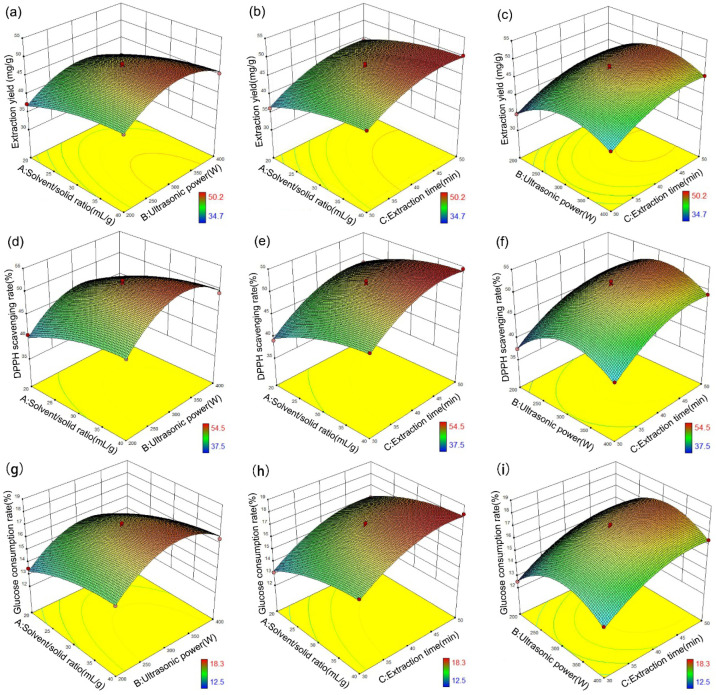
Response surfaces for the interactions of independent variables (A: solvent/solid ratio, B: ultrasonic power, and C: extraction time) on (**a**–**c**) extraction yield, (**d**–**f**) DPPH scavenging rate and (**g**–**i**) glucose consumption rate.

**Table 1 molecules-27-05794-t001:** Components of DESs and physicochemical properties of their aqueous solutions.

DESs	HBA	HBD	Mole Ratio	Viscosity ^a^ (mPa*s)	pH ^a^
DES-1	Choline chloride	DL-Malic acid	1:1	1.621	1.74
DES-2	Choline chloride	L-(+)tartaric acid	2:1	1.732	1.66
DES-3	Choline chloride	Citric acid	1:1	1.843	1.58
DES-4	Choline chloride	DL-Malic acid/proline	1:1:1	1.966	1.61
DES-5	Betaine	DL-Malic acid	1:1	1.586	1.63
DES-6	Betaine	L-(+)tartaric acid	2:1	1.815	1.57
DES-7	Betaine	Citric acid	1:1	1.786	1.82
DES-8	Betaine	DL-Malic acid/proline	1:1:1	1.834	1.72

^a^ Determined in 70 *v*/*v*% aqueous solution at 28 °C.

**Table 2 molecules-27-05794-t002:** BBD matrices with observed values for total yield of flavonoids, DPPH scavenging rate and glucose consumption rate. A: solvent/solid ratio, B: ultrasonic power, and C: extraction time. The concentration of DES-3 solution was 70 *v*/*v*%.

Run	A (mL/g)	B (W)	C (min)	Total Yield of Flavonoids (mg/g)	DPPH Scavenging Rate (%)	Glucose Consumption Rate (%)
1	30	300	40	47.60	51.5	17.2
2	20	400	40	39.64	42.8	14.3
3	30	300	40	47.53	51.4	17.2
4	20	300	50	44.31	48.0	16.0
5	20	200	40	37.35	40.5	13.5
6	20	300	30	36.20	39.1	13.1
7	40	300	50	50.24	54.5	18.3
8	30	400	30	38.20	41.3	13.8
9	30	200	30	34.70	37.5	12.5
10	30	300	40	48.21	52.2	17.4
11	40	200	40	42.35	45.9	15.3
12	30	400	50	45.27	49.0	16.4
13	40	300	30	43.40	47.0	15.7
14	30	200	50	41.72	45.1	15.1
15	40	400	40	45.43	49.2	16.4
16	30	300	40	47.44	51.4	17.2
17	30	300	40	46.72	50.5	16.9

## Data Availability

Not applicable.

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
