# Peer review of "Ultrasound-Assisted Extraction of Flavonoids from Potentilla fruticosa L. Using Natural Deep Eutectic Solvents"

_molecules, 2022, doi:10.3390/molecules27185794_

Round 1
Reviewer 1 Report
The study “Ultrasound-assisted extraction of flavonoids from Potentilla fruticosa L. using nature deep eutectic solvents” fall within the scope of the Molecules Journal.
The MS describes an interesting study, but it needs improvement. Try to improve the methods and discussion which are not very clear. The results must be rewrite in order to easy the understand. The conclusions are not very concluding
Page 1- Line 26: Please specify geographic site
Page 2 - Line 48: Which are the traditional extraction processes?
Page 5 - Lines:190-193: I suggest moving text and table 1 in the material and methods section.
Page 5 - Line:196: Please check (1 or I??)
Rewrite the results in clear way and improve the discussion to elucidate the research findings.
Author Response
Thank you for your review about our paper, we have revised the manuscript according to your comments.
- The specific geographical site has been added in the revised manuscript (Mount Daban in Datong county, Qinghai province).
- The traditional extraction processes have been illustrated in the revised manuscript ( such as cold quenching extraction, thermal reflux extraction).
- The text in lines 190-193 and the table 1 have been moved in the material and methods section based on your suggestion.
- “1”in line 196 was mistakenly written as “I”, which has been modified.
- According to your comments, the results have been rewritten and the discussion section has also been revised. please see the revised manuscript for details.

Reviewer 2 Report
The authors presented a novel method to extract flavonoids from PF by using NADESs with the assistance of ultrasound. The quality of the manuscript is high, and the experiment results can support the conclusions. I would like to recommend this manuscript for publication.
Author Response
We really appreciate your good comments on our paper. Thank you very much.

Reviewer 3 Report
1. As a plant, its generic name and specific epithet should be in italic. So, “Potentilla fruticosa L.” should be “Potentilla fruticosa L.”.
2. Line 32-33, “Flavonoid glycosides mainly include apigenin, quercetin, kaempferol, luteolin, rhamnetin, isorhamnetin, myricetin and so on”. In fact, these seven compounds belong to flavonoid aglycones rather than flavonoid glycosides.
3. Many mistakes were found in English writing. So, this paper should be modified by some professional English editing service.
(1) Line 22, “Nature deep eutectic solvents”.
(2) Line 12, “The morphologies of the raw sample and the samples subjected ultrasonic bath”.
(3) Line 31-32, “it contains a variety of flavonoids, terpenoids and tannins, among which flavonoids are the most typical substances”.
(4) Line 34, “have the effects of lowering blood pressure, lowering blood lipids and antioxidants”.
(5) Line 41, “temperature-sensitive components of the active substances”; Line 43, “active substances”. What were active substances? What was difference between components and active substances?
(6) 99wt%?
(7) Line 92-93, “with analytical pure”.
4. In Equation (1), what are m1 and m0?
5. In “References”, there are many unified formats, such as author names, journal names.
6. In “HPLC analysis”, the gradient elution program was not stated clearly. What was “UV absorbance value of their corresponding peak area”? This quantitative method originated from where? If there was no reference, method validation must be carried out.
7. Line 352-353, “a one-off experiment was performed with 70v/v% DES-3 with a solvent/solid ratio of 40:1 mL/g at 318W ultrasonic power for 47min”. I checked DH-1200E ultrasonic cell crusher (Ning Bo Lawson smart tech Co., Ltd.), which was used in this study. Its single ultrasonic time was 0.1-9.9s and ultrasonic power was 20-1200W (1-99%). So, how did you use it for 47min? How did you set power as 318W?
8. In Table 1, other reagents were written in their English names. So, “CHCl” should be modified as “choline chloride”.
9. The results and conclusions were not stated in “Abstract”.
Author Response
Thank you for your review about our paper, we have revised the manuscript according to your comments.
- We have changed the“Potentilla fruticosa L.” to italic format.
- Thank you for pointing out the error, we have corrected it in the manuscript.
- Thank you for pointing out the mistakes in English writing. In view of the mistakes you pointed out, we have made corresponding modifications.
- m1 is the mass of the flavonoids extracted in the solution and m0 is the mass of PFL powder added. The meanings of M1 and M0 have been restated in the text.
- According to your opinion, the format of author names and journal names in “ References” has been proofread and corrected.
- The gradient elution program was redescribed and the references to quantitative methods was also inserted in the revised manuscript.
- We have checked the model and manufacturer of the ultrasonic cell crusher and found a writing error in the text, which has been corrected now.
- According to your suggestion, “CHCL”in Table 1 has been modified as “choline chloride”
- Appropriate changes have been made into“Abstract” as you suggested.

Reviewer 4 Report
The manuscript "Ultrasound-assisted extraction of flavonoids from Potentilla fruticosa L. using nature deep eutectic solvents" devoted to study of extraction of flavonoids from Potentilla fruticosa L. using mixtures of betaine and choline chloride with organic acids. Solvent/solid ratio, ultrasonic power and extraction time, water content were varied to achieve highest extraction yield, antioxidant capacity and hypoglycemic capacity. The DPPH method and glucose consumption method were used in this study. HPLC method was used to quantitative analysis of various compounds. Single factor method and response surface methodology was used for optimization of extraction conditions. This study is quite interesting for specialists in food chemistry and researchers who deal with NADES.
The manuscript is written clearly, well-structured and has a good scientific soundness. This manuscript can be published in the Molecules journal after minor revision taking into account some of the remarks described below
1. Please, provide more statistical analysis. Effectiveness of various solvents should be compared using Tukey’s test.
Author Response
According to your suggestion, Tukey’s test has been applied to compare the effectiveness of various solvents.

Round 2
Reviewer 1 Report
Page 1, Line 37: The authors should consider the important role of the natural compounds of different species. They could include some relevant and recent citation as for example:
· Bruno, M. R., Russo, D., Cetera, P., Faraone, I., Lo Giudice, V., Milella, L., ... & Gerardin, P. (2020). Chemical analysis and antioxidant properties of orange‐tree (Citrus sinensis L.) biomass extracts obtained via different extraction techniques. Biofuels, Bioproducts and Biorefining, 14(3), 509-520.
· Faraone, I., Russo, D., D’Auria, M., Bruno, M. R., Cetera, P., Todaro, L., & Milella, L. (2020). Influence of thermal modification and extraction techniques on yield, antioxidant capacity and phytochemical profile of chestnut (Castanea sativa Mill.) wood. Holzforschung. https://doi.org/10.1515/hf-2020-0037
· Bioactive compounds achieved from residues of the apricot tree (Prunus armeniaca L.) and olive tree (olea europaea L.): New prospective in the industrial sector
· Bruno, M.R., Cetera, P., Giudice, V.L., Gerardin, P. European Biomass Conference and Exhibition Proceedings, 2021, pp. 186–190
Page 2, Line 53: Check the word “experimenters”
In the conclusions should not summarize the entire work, but reported the main findings of you work and find a link with the future prospects. I suggest another revision.
Author Response
Thank you for your review about our paper, we have revised the manuscript according to your comments.
- According to your suggestion, relevant references have been added.
- Change "Experimenters" to "Experimenter".
- According to your suggestion, the conclusion has been modified appropriately.
